# Methamphetamine Induces Systemic Inflammation and Anxiety: The Role of the Gut–Immune–Brain Axis

**DOI:** 10.3390/ijms231911224

**Published:** 2022-09-23

**Authors:** Majid Davidson, Marina Mayer, Amanda Habib, Niloufar Rashidi, Rhiannon Talia Filippone, Sarah Fraser, Monica D. Prakash, Puspha Sinnayah, Kathy Tangalakis, Michael L. Mathai, Kulmira Nurgali, Vasso Apostolopoulos

**Affiliations:** 1Institute for Health and Sport, Victoria University, Melbourne, VIC 3021, Australia; 2Regenerative Medicine and Stem Cells Program, Australian Institute for Musculoskeletal Science (AIMSS), Melbourne, VIC 3021, Australia; 3Developmental Biology of the Immune System, Life and Medical Sciences (LIMES) Institute, University of Bonn, 53115 Bonn, Germany; 4School of Health and Biomedical Sciences, RMIT University, Melbourne, VIC 3083, Australia; 5First Year College, Victoria University, Melbourne, VIC 3011, Australia; 6Institute for Sustainable Industries and Liveable Cities, Victoria University, Melbourne, VIC 3030, Australia; 7Florey Institute of Neuroscience and Mental Health, Melbourne, VIC 3052, Australia; 8Department of Medicine Western Health, Faculty of Medicine, Dentistry and Health Sciences, University of Melbourne, Melbourne, VIC 3021, Australia; 9Immunology Program, Australian Institute for Musculoskeletal Science (AIMSS), Melbourne, VIC 3021, Australia

**Keywords:** methamphetamine (METH), blood–brain barrier, leaky gut, inflammation, anxiety, gut–brain axis, gut–brain–immune axis

## Abstract

Methamphetamine (METH) is a highly addictive drug abused by millions of users worldwide, thus becoming a global health concern with limited management options. The inefficiency of existing treatment methods has driven research into understanding the mechanisms underlying METH-induced disorders and finding effective treatments. This study aims to understand the complex interactions of the gastrointestinal–immune–nervous systems following an acute METH dose administration as one of the potential underlying molecular mechanisms concentrating on the impact of METH abuse on gut permeability. Findings showed a decreased expression of tight junction proteins ZO-1 and EpCAm in intestinal tissue and the presence of FABP-1 in sera of METH treated mice suggests intestinal wall disruption. The increased presence of CD45+ immune cells in the intestinal wall further confirms gut wall inflammation/disruption. In the brain, the expression of inflammatory markers Ccl2, Cxcl1, IL-1β, TMEM119, and the presence of albumin were higher in METH mice compared to shams, suggesting METH-induced blood–brain barrier disruption. In the spleen, cellular and gene changes are also noted. In addition, mice treated with an acute dose of METH showed anxious behavior in dark and light, open field, and elevated maze tests compared to sham controls. The findings on METH-induced inflammation and anxiety may provide opportunities to develop effective treatments for METH addiction in the future.

## 1. Introduction

Methamphetamine (METH) is a potent psychostimulant and sympathomimetic drug that drastically impacts the user’s body, causing severe neurological and physical consequences. METH use results in increased alertness, heart rate, blood pressure, body temperature, and loss of appetite [1,2]. Chronic METH use commonly leads to severe tooth decay, infection, weight loss, malnutrition, kidney and liver damage, respiratory issues, paranoia, violent behavior, psychosis, severe anxiety, and depression [3,4,5]. METH has a long half-life of approximately 10 h in humans [6], and its stimulatory effects are followed by severe withdrawal, described as an intensely negative emotional state [7]. This often leads to the continued use and can cause a negative reinforcement cycle [8], which may lead to addiction [8]. Even when individuals stop taking METH, the symptoms may persist for years [9,10,11,12,13].

Exposure to METH results in increased intestinal permeability and oxidative stress, inducing gastrointestinal inflammation and ischemia [14,15,16]. It has been reported that bowel ischemia can lead to occluded blood flow to the gut, resulting in the death of intestinal tissue and, in severe cases, organ failure and death [17]. Additionally, disruption of gut wall integrity allows for the leakage of gut contents, including microbiota, toxins, and inflammatory cells, into the circulation [15,18]. Gut-derived components that escape into the circulatory system can travel to the central nervous system (CNS) and cross the blood–brain barrier (BBB), inducing an inflammatory response within the brain [19]. These mechanisms may be the significant contributors to the anxiety and depression often noted in METH users. Neuro-inflammation triggers demyelination and death of dopaminergic and serotonergic neurons within the reward pathway, leading to anxiety and depression [20,21,22].

Excessive adrenergic stimulation following METH exposure has been shown to exert neurocognitive effects [23]. However, recent evidence suggests that neuro-inflammatory processes can also play a significant role in developing neurological disorders in the presence of METH [24,25]. A correlation between METH administration and effects on the expression of tight junction proteins and oxidative stress markers contributing to BBB alterations was shown previously [19,26,27,28]. In addition, METH is associated with increased intestinal permeability; a condition referred to as leaky gut. Therefore, the enteric microbial toxins and activated immune cells can leak from the permeable intestine into the bloodstream. Consequently, endotoxin components can be carried to the brain systemically, causing inflammation and contributing to neuropsychiatric disorders [29,30,31].

This study aims to understand the complex interaction of the gastrointestinal, immune, and nervous systems following an acute METH dose. The findings of this study explain the implications of developing neuropsychiatric disorders, including anxiety following METH administration. Anxiety is one of the major global health concerns, especially amongst the vulnerable group of people, including drug addicts. METH-induced anxiety can lead to suicidal thoughts and self-harm [32,33]. Currently, the only available treatment is via psychological therapy and rehabilitation [34]. However, this treatment often leads to drug relapse. The inefficiency of existing treatment methods has driven research into understanding the mechanisms underlying METH-induced disorders and finding effective treatments [35]. Despite this immense existing knowledge, determining the link between intestinal permeability, BBB integrity, and anxiety development following METH exposure can lead to the development of novel efficient, targeted treatments.

## 2. Results and Discussion

To determine the effects of a single acute dose of METH in mice compared to sham-treated mice, behavioral studies, immunohistochemistry of gut and brain tissues, gene changes in brain tissues, cellular and gene changes in the spleen, and inflammation in the blood were analyzed. A summary of the experimental design is presented in Figure 1, together with a summary of the outcomes.

### 2.1. METH Alters Intestinal Permeability

It is widely acknowledged that METH abuse causes severe damage to the intestine [36,37]. However, the mechanisms underlying METH-induced increases in intestinal permeability and damage to the GI tract leading to neuropsychiatric disorders have not been studied. Therefore, we employed immunohistochemistry in intestinal tissues to understand the influence of METH administration on tight junction proteins compared to sham-treated groups. Figure 2 presents the expression of tight junction protein ZO-1 and EpCAm in colon tissues for METH- and sham-treated mice. METH administration causes a significant reduction in the expression of tight junction proteins within the intestine.

Reduced ZO-1 and EpCAm expression are indicative of tight junction disruption, which leads to increased intestinal permeability and indicates gastrointestinal inflammation [38]. Results of the current study correlate with Mahajan et al. [39] and Sajja et al. [28] findings and confirm that tight-junction proteins, including ZO-1, have an underlying influence on the destructive effects of METH. Furthermore, METH exposure also reduced the expression of the EpCAm. This protein is expressed in tight junctions and adherens junctions of the mouse intestine. It plays a role in cell-to-cell and cell-to-matrix interactions, and in EpCAm knockout mice, there is evidence of impaired intestinal barrier function and dysfunctional ion transport [40]. EpCAm has been shown to disrupt the link between α-catenin and F-actin, resulting in reduced cadherin-mediated cell adhesion [41]. Damage to the intestinal barrier is associated with an increase in permeability between the external gut environment and the internal vasculature of the body, a phenomenon commonly known as leaky gut. Disruption of the gut wall integrity, damage to intestinal epithelial cells, and derangement of tight junctions lead to the leakage of microbial products, macromolecules, and microbiota from the intestinal lumen into the blood circulation, liver, spleen, and mesenteric lymph nodes, which in turn causes peripheral inflammation [18,42].

### 2.2. METH Increases Fatty Acid Binding Protein 1 in Blood

Fatty acid binding protein 1 (FABP-1; also known as liver-type FABP) is highly expressed in the liver, and in the epithelial mucosal layer of small intestines. FABP is present in mature epithelial villi in the intestine and is released into the circulation from enterocytes upon damage to the mucosal epithelial layer [43]. Mice fed on a high-fat diet also show increased concentrations of FABP-1 in the circulation, and damage to the mucosal layer is hypothesized to be due to obesity [44]. The presence of gut-derived FABP-1 in the serum is indicative of increased gut-wall permeability [45]. It was shown that FABP-1 levels in the serum for acute METH group were higher than in sham-treated mice (Figure 2D). Therefore, results revealed that administration of METH causes intestinal mucosal layer damage, which leads to increased concentration of FABP-1 in the circulation. METH causes increased levels of norepinephrine release and thus causes vasodilation in blood vessels in the brain whilst vasoconstriction in the blood vessels going to the gut, causing constipation, bowel ischemia, and abdominal cramping [15]. Upon prolonged abuse, this vasoconstriction may cause damage to the intestinal cells, thus causing increased permeability.

### 2.3. METH Alters Cellular Immune Responses in the Colon

The effects of METH exposure on gastrointestinal inflammation were assessed via immunohistochemistry on colon sections using the pan-leukocyte marker, anti-CD45 antibody. At the same time, the DAPI (blue) highlights the nucleus of all cells. Colon tissues displayed a significant increase in the expression of CD45+ cells because of acute METH administration (Figure 3), indicating an inflammatory reaction within the gut. This study is one of the few to investigate the immune effects of acute METH exposure in the colon. The colon is an essential part of the large intestine, which serves powerful functions, including absorption of water, electrolytes, remaining nutrients, and movement of feces towards the rectum for elimination [46,47,48]. An inflamed colon is associated with many disorders, particularly ulcerative colitis, an inflammatory bowel disorder characterized by inflammation and ulcers within the innermost lining of the digestive tract [49].

### 2.4. METH Increases BBB Permeability

The BBB is a highly selective interface that protects the brain from many of the constituents of the peripheral circulation. It allows the passage of gases, water, and lipid-soluble molecules (including METH) from the CNS in a healthy state, yet, it prevents unwanted molecules such as toxins, circulating immune cells, and bacteria [50]. METH increases BBB permeability, inducing damage by altering the structure of proteins involved in BBB stability [51]. METH alters BBB permeability via dysregulation of the tight junction proteins, including occludin and ZO proteins [39,52]. In vitro studies using a single dose of METH (1µM) on endothelial monocultures showed increased permeability within less than 1 h [27]. Moreover, a previous study by Matines et al. (2011) reported that a single dose of METH (30 mg/kg) in mice leads to a peak plasma concentration after about 1 h [53]. Administration of METH at such concentrations in rodents consistently leads to BBB breakdown. In mice brain, significant accumulation of plasma proteins (albumin or IgG) was observed after several hours [54]. In a similar study by Bowyer et al. (2008), findings showed that administration of a single dose of METH (40 mg/kg) in mice induced BBB changes for 1.5 h to 3 days [55]. In our study we used a single dose of METH (30 mg/kg), which induced increased BBB permeability evidenced by the presence of gut-derived albumin in the brain tissue.

Albumin is a large protein and does not cross the BBB, and its presence in the brain is a commonly accepted indicator of disruption to the BBB [56]. Immunohistochemistry analysis of brain hippocampus tissues showed an increase in albumin expression in METH-mice compared to sham-treated controls (Figure 4). Findings are consistent with previous reports demonstrating an association between METH exposure and increased permeability of the BBB in brain regions, including the hippocampus [53,57,58]. Damage to the BBB can permit an influx of bacteria, pro-inflammatory mediators, and peripheral immune cells directly from the circulation.

### 2.5. METH Causes Gliosis

The anti-glial fibrillary acidic protein (GFAP) antibody was used to label astrocytes, and an anti-TMEM119 antibody was used to detect microglial cells. GFAP is an intermediate filament protein expressed by astrocytes, and its upregulation is used to determine astrogliosis within the CNS. No significant differences in GFAP expression were noted between the sham-treated and the METH-treated groups, as the astrocytes did not appear to change due to an acute METH dose. However, there was a significant increase in the expression of TMEM119 protein in the brain hippocampus of METH-treated mice. TMEM119 (a type-1 transmembrane protein) is a marker expressed explicitly by microglial cells, and its upregulation can indicate microgliosis and inflammation in the brain (Figure 5). Our results showed no significant increase in GFAP expression 3 h after METH administration. This finding is opposite to previously reported results for in vitro studies of rat fetal mesencephalic cell lines where METH increased GFAP expression, which was inhibited by benzamide, an inhibitor of ADP-ribosylation [59]. These differences might be because of differences between in vivo and in vitro models. Furthermore, Rats treated with 10 mg/kg of METH and microglia assessed 2 h and 3 days post-administration showed gene transcription changes (cyclo-oxygenase 2, prostaglandin E2, glutamine uptake, and nuclear factor eryhroid 2-related factor 2) in the rat striatum and prefrontal cortex at 2 h and not 3 days [60]. However, in human brain tissues from METH users who died from intoxication, GFAP, and S100B expression were not significantly increased, although expression of hGLUT5 was significantly increased [61]. Our studies show that a single dose of METH alters the expression of TMEM119, a marker for microglial cells and inflammation.

### 2.6. METH Administration Changes Gene Expression in the Mid-Brain

Reverse transcription-qualitative polymerase chain reaction (RT-qPCR) was used to determine the expression of pro-inflammatory cytokines (IL-1β, TNFα, and IL-6), chemokines (Cxcl1 and Cxcl5), and glial cell markers (GFAP, Iba-1, and S100b) in the brain sample. In the brain, significant increases were noted at the mRNA expression levels of Ccl-2 (*p* < 0.005), IL-1β (*p* < 0.01), TNFα (*p* < 0.05) and Cxcl1 (*p* < 0.05) in the METH-treated group compared to the sham group (Figure 6). In addition, a significant increase in Ccl-2 (MCP-1) and IL-1β expression were also noted in the hippocampus (Figure 7). IL-6 (inflammatory marker), Cxcl5 (chemokine receptor), GFAP (astrocytes), S100b (microglial cells), and lba-1 (microglial cells) mRNA expression were not detectable in either the brain or hippocampus following METH administration. In contrast with some reports, increased transcription of the pro-inflammatory cytokines IL-6 has been shown [62]. Nevertheless, it is clear from previous observations that METH effects in the brain are region/time/dose-dependent. The previous study showed that an acute high dose of METH (30 mg/kg) induces an early increase in the expression levels of IL-6 mRNA in the hippocampus, frontal cortex, and striatum, and TNFα mRNA only in the hippocampus and frontal cortex [62]. Furthermore, a similar study evaluated the effect of a single doses of METH (0.5, 1, 2, and 4 mg/kg) 2 h apart in rat model [63]. Findings revealed that METH had a dose-dependent stimulatory effect on locomotor activity over the 8 h. A significant increase in dopamine concentration was reported in the frontal cortex with the highest dose of METH 2 h after the dose administration [63]. This effect was dose- and region-specific, as no significant changes was reported for lower doses, nor was a significant change reported for other brain regions.

Several studies have shown that glial cells (including microglia and astrocytes) participate in METH-induced neuroinflammation in the brain [64,65]. Microglial cells are the resident innate immune cells of the CNS. In a resting state, microglial cells continuously assess the environment for damage, the presence of foreign bodies, including pathogens, and markers of inflammation, including DAMPs [65,66]. Once activated, microglia play an integral role in the direct response to injury through their function in the innate immune system. They do this in several ways, including the release of pro-inflammatory cytokines, cytotoxic substances, and phagocytosis. Initially, the pro-inflammatory state assumed by microglial cells was assumed to be a consequence of the neurotoxicity and neuronal damage caused by METH administration. However, more recently it has been speculated that the reactive microglia may contribute to the complex mechanism of neuronal damage following METH exposure by producing pro-inflammatory mediators. Although a significant increase was not observed in the classical microglial marker Iba-1 or astrocyte markers S100b and GFAP, increased expression of the pro-inflammatory cytokines, including IL-1β, is a recognized sequel following glial activation, further implicating glial cells in the neuroinflammation caused by METH. It is noteworthy that changes in proliferation or migration of glial cells into areas of damage, also associated with the activation process, are not expected to be detectable using RT-qPCR.

Furthermore, Ccl-2 and IL-1β are recognized as essential mediators of neuro-inflammation and have identified roles in trafficking to the brain. Ccl2 is a chemokine that shows chemotactic activity for monocytes and Cxcl1 for neutrophils. In addition, a chemo-attractant for neutrophils in the brain is consistent with increased recruitment and leukocyte infiltration of CNS in response to METH.

### 2.7. METH Alters Cellular Immune Responses and Genes in the Spleen

Flow cytometry analysis was used to determine the effects of acute METH administration on splenic immune cell populations. METH exposure caused transient immunosuppression characterized by a decrease in the spleen’s proportion of granulocytes, NK, and CD4+ T cells at time t = 3 h (Figure 8A); at time t = 24 h the cell populations were back to baseline levels, hence transient effect. The effects of METH on the spleen are mainly unknown. The spleen acts as a significant reservoir of immune cells that can be mobilized into the circulation and recruited to the sites of injury and inflammation under chemokine gradients. Splenic-derived immune cells have been shown to contribute to, and possibly exacerbate, neuroinflammation in several brain injuries, including stroke [67] and head trauma [68,69,70]. In the case of stroke, it has been shown that splenic neutrophils are recruited within 24 h, and these cells play an essential role in BBB destabilization by releasing MMP-9, which in turn allows more leukocyte infiltration increasing neuroinflammation [71]. A decrease in cell populations does not necessarily reflect immunosuppression but might instead be a product of mobilization to other parts of the body. Consistent with previous reports, acute METH caused a transient reduction in the proportion of granulocytes, NK cells, and T-cells in the spleen [72]. Whether this reflects altered trafficking from the spleen in response to METH-induced peripheral inflammation or reflects aspects of immunosuppression is not clear.

Using RT^2^ Profiler PCR gene arrays showed that Ccl-2 (MCP-1), Cxcl1, Cxcl5 and TNFα mRNA were expressed at significantly lower levels following METH administration (Figure 8B). Downregulation of Ccl-2 and TNFα was also confirmed by RT-qPCR using individual cDNAs (not shown). It is assumed that this downregulation is the product of pro-inflammatory cytokine TNFα and NF-κβ activation, which is elevated in the brain by METH [73,74]. The upregulation of cell adhesion molecule gene expression has been observed in the brain of METH abusers, which plays a neuroprotective role in improving the tightness of the BBB [75]. The results of the array also showed up-regulation of IL10 and Aicda genes (Figure 8B).

### 2.8. METH Causes Behavioral Changes

Animal models are valuable and standard methods for studying human diseases [76]. Spontaneous activity in rodents is frequently used to study the neurobiology of anxiety [77] and major depressive disorder [78,79]. For this purpose, several test models have been developed [80], including open field test (OFT), light and dark test (LDT) and elevated maze test (EMT) for anxiety [81].

Herein, three established behavioral tests, including LDT, OFT, and EMT, were performed to assess anxiety levels following METH vs. sham administration (Figure 9). The LDT is an anxiety-like behavior test that places mice into a conflict situation between their drive to explore new unknown areas and their aversion to being in brightly illuminated open spaces [82]. The open-field test combines stress factors such as separation from the other mice and fear of the large unprotected bright surface. Therefore, it is appropriate to investigate anxiety and behavior toward new unknown environment and observe general locomotor behavior [83]. The EMT is designed to observe the state of anxiety in rodents by using a plus-shaped maze that combines the natural antipathy of mice towards open elevated environments with their instinct to discover new environment [84]. These tests provided parameters related to general movements, exploratory and locomotor behavior that were employed to estimate the level of anxiety in mice following METH administration or sham treatment.

The LDT was performed as the first test. LDT is based on the natural aversion of mice to the open and bright spaces compared to dark spaces and their spontaneous exploratory behavior in response to mild stressors, such as novel environment and light. Figure 9A shows the movement of mice in the lightbox 3 h following administration of an acute dose of METH and sham. The METH-treated mice demonstrated significantly lower activity in the light box (0.7 ± 0.2 m) compared to the sham group (2.1 ± 0.3 m). Results reveal that the METH-treated group spent less time (2.1 ± 0.2 min) in the dark box compared to the sham group (5.2 ± 0.4 min). According to the data obtained in LDT, the time spent in the lightbox for METH treated group was more than the sham-treated group. Additionally, METH-treated groups had less movement (distance travel) in the lightbox compared to the sham group. Moreover, the number of transitions between the two compartments of the light/dark box was significantly decreased for the METH treated group compared to the sham group.

OFT was employed as the second behavioral study to evaluate the level of anxiety of treated mice. Figure 9B shows the travel distance map for METH and Sham treated groups. Results show that the METH treated group spent more time near the corner (4.1 ± 0.2 min) compared to the sham-treated group (2.9 ± 0.1 min). Furthermore, the METH treated group spent less time (0.1 ± 0.1) in the middle of the box compared to the Sham group (1.3 ± 0.2 min). In addition, other parameters like immobility (freezing), average speed, and the total distance showed significant differences between METH and Sham treated groups. The reduction of time spent in the center and the increased time spent in the corner is strongly associated with higher anxiety levels, as the mice fear open and unprotected areas [85]. Anxious mice avoid the center and open area and spend more time in the corners, which gives them more shelter [86]. The METH-treated animals showed a decrease in total distance travelled and a reduction in exploratory and locomotor behavior (as indicated by the transitions between the different areas in the open field), indicating a higher level of anxiety [85].

The EMT was performed as the third behavioral study to evaluate the level of anxiety for each group. Results show that the METH-treated group spent more time (2.7 ± 1.1 min) in the closed arms compared to sham-treated (2.1 ± 0.6 min) and less time (1.2 ± 0.5 min) in the open arms compared to the sham-treated group (2.4 ± 0.1 min) (Figure 9C). In addition, the number of the entrance to closed arms was significantly lower for the METH-treated group (5 ± 0.5) compared to the sham-treated group (14 ± 3, *p* ≤ 0.05). Moreover, the entries/transitions into open and closed arms were lower for the METH-treated group indicated also decrease of locomotor activity. High anxiety levels have been associated with an increase in time spent in the closed arms of the elevated maze and a decrease in the number of entries into either of the open/closed arms. This result may be due to indifferences to surroundings and decreased exploratory behavior and general movement [87]. Therefore, METH-treated animals have a higher level of anxiety than the sham-treated group.

## 3. Materials and Methods

### 3.1. Animals

All experimental procedures and animal care were approved by the Victoria University Animal Experimentation Ethics Committee and carried out following the guidelines of the National Health and Medical Research Council Australian Code of Practice for the Care of Animals for Scientific Purposes. Female C57BL/6 mice aged 8 weeks (18–22 g) purchased from the Animal Resource Centre (Perth, Australia) were used for the experiments. Mice had free access to food and water and were kept under a 12 h light/dark cycle in a well-ventilated room at an approximate temperature of 22 °C. Mice acclimatized for a minimum of 5 days prior to the commencement of experiments.

### 3.2. Treatment Protocols

Mice were randomly assigned to either METH- or sham-treatment groups. Methamphetamine hydrochloride (>98%, National Measurement Institute, Lindfield, NSW, Australia) (30 mg/kg body weight) was diluted in injectable sterile saline. All working solutions were prepared separately based on animal body weight to ensure that all animals received the same dose and 0.1 mL was administered via a single intraperitoneal injection. This study followed the simple practice guide for dose conversion between animals and humans [88] to calculate the METH dose similar to the drug levels in human users. Previous studies showed a single dose of METH induced neurotoxicity in rodents [89]. The intraperitoneal injection was chosen due to the ability to control the dosage amount accurately, and the administration is systemic, so the METH will reach the blood with complete bioavailability. A previous study by Harris et al. showed the elimination half-life of METH for intravenous and intranasal were 11.4 and 10.7 h, respectively [6]. In a similar study, Hendrickson et al. showed that the route of administration did not significantly affect these pharmacokinetic parameters of METH [90]. Sham-control mice received a single injection of 0.1 mL 0.9% sodium chloride solution. The maximum injection volume did not exceed 200 µL per injection for all mice. A single dose of METH injection corresponds to acute-METH administration protocol.

### 3.3. Behavioral Assessment

Three established behavioral tests, including LDT, OFT, and EMT, were performed to assess anxiety levels following METH- or sham- exposure. The behavioral studies were performed after three hours of treatment administration. Mice were acclimatized in their cages in the procedure room for 1 h before commencing behavioral studies. All tests were conducted in the same order due to the consistency of the experimental design. The experimental time for all animals was identical, exactly 3 h after METH administration.

#### 3.3.1. Light and Dark Test (LDT)

This model of anxiety was based on the previously described method by Misslin et al. (1989) [91]. The chamber was made of polyvinyl chloride covered with plexiglass. The lightbox (28 × 20 × 20 cm) is larger than the dark box (15 × 20 × 20 cm). These boxes are attached on one side with a cut-out at the bottom (5 × 5 cm) and a sliding door with which the channel can be opened and closed. The light intensity in the center of the illuminated box was 900 lux, while the interior area of the dark box was utterly dark [83]. At the beginning of the experiment, a mouse was placed in the illuminated box. Then, the sliding door was removed after five seconds, and the mouse could explore freely between the chambers. A bird’s eye view camera captured the entire test session, and the following parameters during a 10 min period were analyzed automatically by an in-housed-developed software: (a) travel distance; (b) light/dark boxes transition, and (c) time spent in the dark box. Although the test-induced anxiety may contaminate these parameters, they were recorded to measure general motor activity in the same context as the anxiety evaluation. The results were expressed as a mean of total traveled distance in the lightbox, mean total number of light/dark box transitions, and mean time spent in the dark box. The software automatically generated a tracking map based on the mouse travel road map in the lightbox. The travel map was used for quantitative comparisons between treatment groups.

#### 3.3.2. Open-Field Test (OFT)

The test employed a square box of white matt polyvinyl chloride with internal dimensions of 72 × 72 cm and 33 cm height of walls [92]. The open box was illuminated with approximately 900 lux during the experiment. At the start of each test, the mouse was placed in the middle of the box and allowed to explore the box over the test period. A bird’s eye view camera captured the entire test session, and the following parameters during a 5 min period were analyzed automatically by an in-house-developed software: (a) travel distance; (b) Time spent in corners and center (Software was programmed to monitor 24 × 24 cm center of the arena), (c) Transition in corners and center and (d) Immobility (grooming) period. The results were expressed as mean total travel distance in the box, meaning a total number of center and corners transition, mean time spent in the corner and centers, and meantime of immobility (grooming) period. The software automatically generated a tracking map based on the mouse travel road map in the box. The travel map was used for quantitative comparison between treatment groups.

#### 3.3.3. Elevated-Maze Test (EMT)

The maze was elevated to a height of 45 cm with two enclosed (5 × 30 cm) and two open arms (5 × 30 cm), arranged so that the arms of the same type were opposite each other, connected by an open central area (5 × 5 cm). To prevent mice from falling off, a rim (0.5 cm high) surrounded the perimeter of the open arms [93]. The light intensity on the central platform was 900 lux. At the beginning of the study, each mouse was placed on the middle platform of the cross, facing one of the open arms, and allowed to explore the cross over the test period. A bird’s eye view camera captured the entire test session, and the following parameters during a 5 min period were analyzed automatically by an in-housed-developed software: (a) time spent in arms and center and (b) transition between arms. The results were expressed as the mean time spent in arms and center and the mean total number of transitions between the corner and centers.

### 3.4. Blood and Tissue Collection

Following the behavior studies, mice were euthanized for tissue collection immediately with an administration of 200 µL pentobarbital (80 mg/kg body weight). First, the cardiac puncture technique was used to collect blood from the heart. Then, distal colon tissues were collected, opened along the mesenteric border, and pinned flat with the mucosa facing upwards onto a Krebs solution-filled Sylgard-lined Petri dish. The tissue was fixed overnight at 4 °C with Zamboni’s fixative (2% formaldehyde containing 0.2% saturated picric acid). The next day the tissues were washed with dimethyl sulfoxide (DMSO) (Sigma-Aldrich, Macquarie Park, NSW, Australia) (3 × 10 min) followed by PBS (3 × 10 min). After washing, tissues were embedded in optimal cutting temperature compound (OCT-compound, Tissue-Tek, St, Torrance, CA, USA) and frozen using liquid-nitrogen and 2-methyl butane (Sigma-Aldrich, Australia) and stored in a −80 °C freezer as in our previous studies described [94]. In addition, the brain perfusion fixation technique was employed on brain tissue to remove blood proteins that can interact with antibodies in the ex vivo studies.

### 3.5. Immunohistocemistry

#### 3.5.1. Gut Tissue

Immunohistochemistry (IHC) was used to evaluate changes in immune cell infiltration and gut permeability. Therefore, the expression of the immune cells, labeled with a pan-leukocyte marker (anti-CD45), tight junction proteins, labeled with a zonula occludents-1 marker (anti-ZO-1), and epithelial cells, labeled with an epithelial cell adhesion molecule marker (anti-EpCAm) in colon tissues were studied following METH- and sham-administration. Tissues were cut into 20 µm sections using a Leica CM1950 cryostat (Leica Biosystem, Nussloch, Germany), adhered to slides, and allowed to rest for 30 min at room temperature before processing. The slides were treated with 10 % (*v*:*v*) normal donkey serum (Merck Millipore, Darmstadt, Germany) and incubated for one hour at room temperature. Sections were labeled with EpCAM (Rabbit, 1:750, Abcam, Cambridge, UK), ZO-1 (Rat, 1:500, Invitrogen, Waltham, MA, USA), and CD45 (Rat, 1:500, Abcam, UK) antibodies overnight at room temperature. Slides were washed and labeled with secondary antibodies (Jackson ImmunoResearch, West Grove, PA, USA) for 2 h at room temperature. Finally, 4′,6′-diamino-2-phenylindole dihydrochloride (DAPI; blue) was used to highlight the nucleus of all cells. All tissues were mounted on glass slides with a fluorescent mounting medium (DAKO, North Sydney, NSW, Australia).

#### 3.5.2. Mid-Brain Tissue

IHC was used to evaluate the BBB permeability changes in the brain’s hippocampus region in sham vs. METH-treated groups. Immune cells were labeled with the pan-leukocyte marker (anti-CD45), gut-derived albumin was labeled with an anti-albumin antibody, astrocytes were labeled with an anti-glial fibrillary acidic protein (GFAP) antibody, and microglial cells were labeled with anti-TMEM119 antibody. Brain samples were cryo-sectioned towards sagittal (level 10–15 using 15 µm intervals) and coronal regions of the hippocampus (bregma −1.46 mm to −2.30 mm using 20 µm thickness). Tissues were mounted on glass slides for IHC and histological techniques. Prepared slides were treated with Zambonis’ fixative (2% formaldehyde and 0.2% saturated picric acid) for 105 min at room temperature. Specimens were subjected to a 1× PBS wash for 10 min followed by 10 min incubation with hydrogen peroxide to block endogenous peroxidase activity. Heat-Induced epitope retrieval (HIER) was used to expose epitope biding sites to ensure selected antibody conjugation. Slides were submerged in 1 L of 0.01 M of sodium citrate for 20 min and maintained at 85 °C, followed by a cooling period for a further 20 min before removing from the buffer. Following HIER, samples were washed in 1× PBS+ Triton-X (0.1%, 1×PBS-T) 3 times for 10 min. They were incubated with 10% normal donkey serum (NDS) (Merck Millipore, Australia) at room temperature for 1 h prior to labeling with primary antibodies. Primary antibodies (GFAP; goat, 1:1000; TMEM119; rabbit, 1:1000; Albumin; rabbit, 1:500; CD45, rat, 1:1000) were applied to each slide in 200 µL PBS-T (2% NDS), saturating the sections overnight at room temperature. Sections were washed in 1× PBS-T (3 × 10 min) and incubated with fluorophore-conjugated secondary antibodies (Abcam). All specimens were labeled with DAPI to label cell nuclei before mounting onto glass slides with DAKO.

Images were captured using a Nikon Eclipse Ti laser scanning microscope (Nikon, Tokyo, Japan). Three-dimensional images were captured at 40× magnification so that eight random images covered a total area of 0.4 mm^2^. Images were analyzed using ImageJ (the original ImageJ/ImageJ 1.x). Images were converted from RGB to 8-bit, after which the auto threshold method of Max Entropy was applied to all images. Once the threshold was set for each image, the number of pixels of positive staining per field of view (FOV) was counted using the Analyze Particles function by the original ImageJ/ImageJ 1.x.

### 3.6. Fatty Acid Binding Protein-1 Assay

The small intestine is responsible for the assimilation of dietary lipid as well as the reuptake of bile acids via the enterohepatic circulation. Differentiated enterocytes of the intestine express high levels of two gut-derived Fatty Acid Binding Proteins (FABPs): FABP1/LFABP and IFABP/FABP2. The distal small intestine expresses a third member of the FABP family, ileal bile acid-binding protein (ILBP; FABP6) [95]. The presence of gut-derived FABP in the serum is considered indicative of increased gut-wall permeability [45]. FABP-1 levels in the serum were measured in METH- and sham-treatment groups using Quantikine ELISA (mouse/rat FABP1/LFABP) (R&D Systems, in vitro Technologies, Noble Park North Victoria, Australia).

### 3.7. RNA Isolation

Tissues (spleen, brain (the brain tissue after removing prefrontal cortex (PFC) area, cerebellum, and brainstem), and hippocampus) were snap-frozen and stored at −80 °C for gene expression analysis. Total RNA was extracted using TRIzol™ (Invitrogen) and purified using RNeasy^®^ Mini kit (Qiagen, Chadstone, VIC, Australia), including DNase treatment. Initial homogenization of the tissue was performed using TissueLyser (Qiagen). The integrity of the total RNA samples was assessed on an Agilent 2100 Bioanalyzer (Agilent Biotechnologies, Santa Clara, CA, USA) with an RNA 6000 Nano Kit (Agilent Biotechnologies, Santa Clara, California, USA), wherein an RNA Integrity Number (RIN) of more than 8.5 was set as the prerequisite for subsequent gene expression analysis. The concentration of individual RNA samples was measured using a Qubit RNA BR Assay (Invitrogen, MA, USA). The purity of the RNA was determined by measuring the 260/280 and 260/230 ratios using Denovix DS-11 Spectrophotometer (Gene Target Solutions, Sydney, Australia).

### 3.8. RT^2^ Profiler PCR Gene Arrays

The RT^2^ Profiler PCR array *Mouse Cancer Inflammation and Immunity Crosstalk* (Qiagen) was used to determine the expression of selected immune and inflammation-related genes in the spleen. This gene array was chosen as it includes a selection of inflammatory and immune-related genes. Reverse transcription was carried out with the RT^2^ First Strand kit (Qiagen, Chadstone, VIC, Australia) using 0.5 µg pooled RNA as a template. PCR was performed in a Biorad CFX96 thermal cycler using the recommended PCR cycling conditions, followed by melt curve analysis to verify PCR specificity. Data analysis was performed using Bio-rad CFX Maestro™ software, version 1.1, normalizing expression to the mean of five reference genes (*Gapdh*; *B2m*; *Actb*; *Gusb* and *Hsp90ab1*). Fold change was determined using the *delta-delta CT method* [96]. The arrays were performed in duplicate, and specific genes showing an expression difference of ≥2-fold relative to control were selected for validation using gene-specific RT-qPCR. Low genes expressed at levels near the detection limits (Ct > 35) and genes without a unique melting point were not considered further.

### 3.9. RT-PCR

RT-PCR was used to determine the expression of pro-inflammatory cytokines (IL-1β, TNFα, and IL-6), chemokines (Cxcl1 and Cxcl5), and glial cell markers (GFAP, Iba-1, and S100b) in the mouse brain (Hippocampus and brain (the brain tissue after removing prefrontal cortex (PFC) area, cerebellum, and brainstem). RT-PCR was performed with cDNA reverse transcribed from DNase-treated RNA (1 µg) using SuperScript IV VILO Master Mix (Life Technologies, ThermoFisher, Melbourne, Australia). PCR was performed using SsoAdvanced™ SYBR Green Supermix (Bio-Rad, CA, USA) with pre-designed primer pairs from Integrated DNA Technologies (IDT, Coralville, IA, USA). All primer sets were intron spanning. PCR reactions included a minimum of two technical replicates for each biological replicate. Data analysis was performed using CFX™ Maestro (Bio-Rad), version 1.1). Normalized relative quantities (NRQ) were determined using the formula 2^−delta−Ct^, where delta-Ct represents the difference between the METH- and sham-treated groups, normalized to the mean of stable reference genes (hippocampus *Actin, Gus, Gapdh; mid- brain Gapdh, Gus*). Data are presented as fold-change, compared to non-treated controls.

### 3.10. Statistical Analysis

Analyses were performed using Graph Pad Prism (Graph Pad Software Inc., San Diego, CA, USA) software and applied Student’s unpaired two-tailed *t*-test or ANOVA to compare data between multiple groups. Statistical analysis of RT-PCR significance was determined using an unpaired, two-tailed *t*-test on log_2_ transformed normalized relative expression (NRQ) values. All data are presented as mean ± standard error of the mean (SEM).

## 4. Conclusions and Future Directions

METH is a member of the amphetamine class of drugs, a group of highly addictive synthetic psychoactive stimulants that cause neurotoxic effects in the CNS [97]. In recent years, METH use has increased dramatically, becoming a serious public health problem worldwide [98,99,100,101]. Currently available treatments for METH addiction and its neuropsychiatric disorders are inadequate and associated with many side effects. The inefficiency of the existing treatments requires the development of novel, more effective therapies for METH addiction, which is a growing public health concern. The inefficiency of existing treatment methods has driven research into understanding the mechanisms underlying METH-induced disorders and finding effective treatments. One of the potential underlying molecular mechanisms focusses on the impact of METH abuse on gut permeability. METH causes rapid and sustained release of stress mediator, norepinephrine, which results in arterial vasoconstriction leading to increased heart rate and hypertension. Similar effects can also be seen in the mesenteric vessels leading to acute intestinal ischemia [14,102]. In METH users, the most common effects of gastrointestinal (GI) vasoconstriction and bowel ischemia include abdominal or stomach cramping, constipation, diarrhoea, and tissue dehydration. In some cases, loss of blood flow to GI muscles leads to severe, potentially fatal GI problems such as paralytic ileus [103]. Potential consequences of paralytic ileus include severe infection, tissue death (gangrene), formation of holes in the intestinal wall and serious disruptions in the levels of electrolytes. In severe cases bowel infarction can lead to development of septic shock with multiple organ failure [102]. Bowel ischemia associates with increased intestinal permeability. Several findings suggest that dysfunction of intestinal mucosal barrier leading to increased epithelial permeability and systemic inflammation which plays an important role in the pathophysiology of depression, cognitive decline, anxiety, chronic fatigue, and eating and sleep disorders. All of these are present in METH users. However, the link between METH-induced increase in intestinal permeability and damage to the GI tract leading to systemic immune response and neuropsychiatric disorders has not been elucidated. This study aimed to understand the complex interactions of the gastrointestinal-immune-nervous systems following an acute METH dose administration.

The findings of this study showed that the expression of tight junction proteins ZO-1 and EpCAm in intestinal tissue has significantly decreased after single dose of METH administration. Moreover, the presence of FABP-1 in sera of METH treated mice suggests intestinal wall disruption. The increase in expression of CD45+ immune cells in the intestinal wall further confirms gut wall inflammation/disruption. Several findings suggest that dysfunction of intestinal mucosal barrier leading to increased epithelial permeability and systemic inflammation which plays an important role in the pathophysiology of depression, cognitive decline, anxiety, chronic fatigue, and eating and sleep disorders. In addition, findings revealed that in the brain, the expression of inflammatory markers Ccl2, Cxcl1, IL-1β, TMEM119, and the presence of albumin were higher in METH mice compared to shams, suggesting METH-induced blood–brain barrier disruption. METH decreases BBB structural proteins and increases BBB permeability to various molecules and gut-derived endotoxins into the brain. In this study presence of gut-derived albumin was used to confirm increases in the BBB permeability. However, the levels of neurofilament light protein (NFL), a cytoskeletal polypeptide of the axon, and chitinase 3-like 1 (CHI3L1, also known as YKL40 or gp39), a glycoprotein secreted by activated glia in the CNS, could be considered to confirm BBB leakage in future studies. Moreover, findings showed the cellular and gene changes in the spleen 3 h after METH administration, indicating reduced immune cell populations in the spleen. This might be due to the migration of immune cells from the spleen to the gut and induction of peripheral inflammation in response to a single acute METH dose administration. Finally, the behavioural study results indicate that the mice treated with an acute dose of METH showed anxious behavior compared to sham controls.

It is well-established that METH can lead to various neuropsychiatric issues, including cognitive decline, anxiety, depression, violent behavior, and psychosis. However, the complex mechanisms underpinning these effects are not yet fully understood. Due to its lipophilic nature and structural similarity to dopamine, METH quickly penetrates the CNS exerting a range of pharmacological effects directly on the nerve cells and neurotransmitter function. METH also exerts significant pro-inflammatory effects [72,104,105,106,107]. This study determined the cellular and molecular changes in the gut, brain, and spleen concurrently to elucidate further the neuro-immune interactions that might predispose the development of METH-induced mental health issues. Moreover, METH-induced alterations in the gut microbiome increase intestinal permeability, and intestinal inflammation may be correlated with the observed CNS alterations leading to METH-induced anxiety and depression. Therefore, further study is required to evaluate the association of METH-induced anxiety and depression and gut microbiota or its metabolites.

In conclusion, we demonstrated that a single dose of METH induces complex interactions of the gastrointestinal-immune-nervous systems. Understanding the mechanisms underlying METH-induced inflammation and anxiety might provide an opportunity to develop an effective treatment for METH addiction in the future. The findings of this study suggest several potential therapeutic pathways that might lead to designing an efficient treatment against METH-induced anxiety and depression. These options include but are not limited to (i) targeting leaky gut to prevent entrance of endotoxins to the blood circulation and brain, (ii) reducing BBB disruption, (iii) immunomodulation to prevent inflammation. Studies are under way in our laboratory to understand the mechanisms involving METH-induced disorders and design an effective treatment against METH addiction [35].

## Figures and Tables

**Figure 1 ijms-23-11224-f001:**
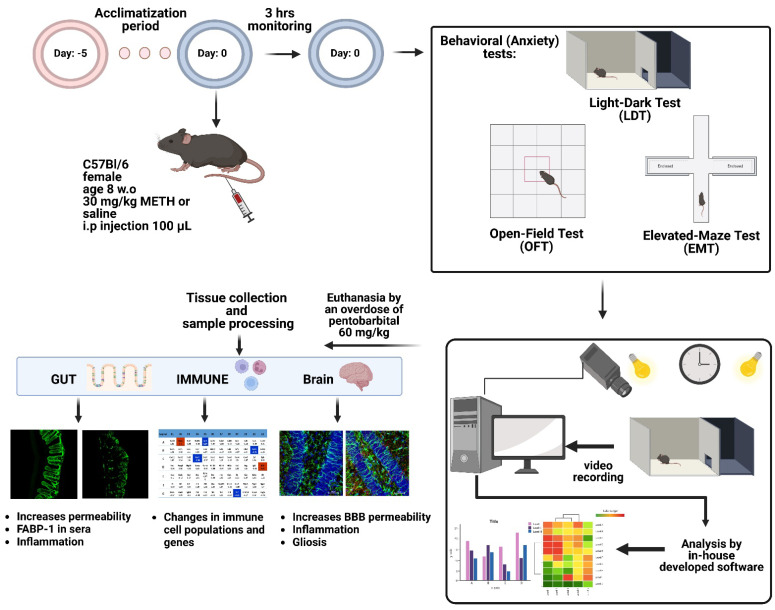
Schematic summary of the experimental design and outcomes in the gut, brain, spleen and sera of mice following an acute dose of METH.

**Figure 2 ijms-23-11224-f002:**
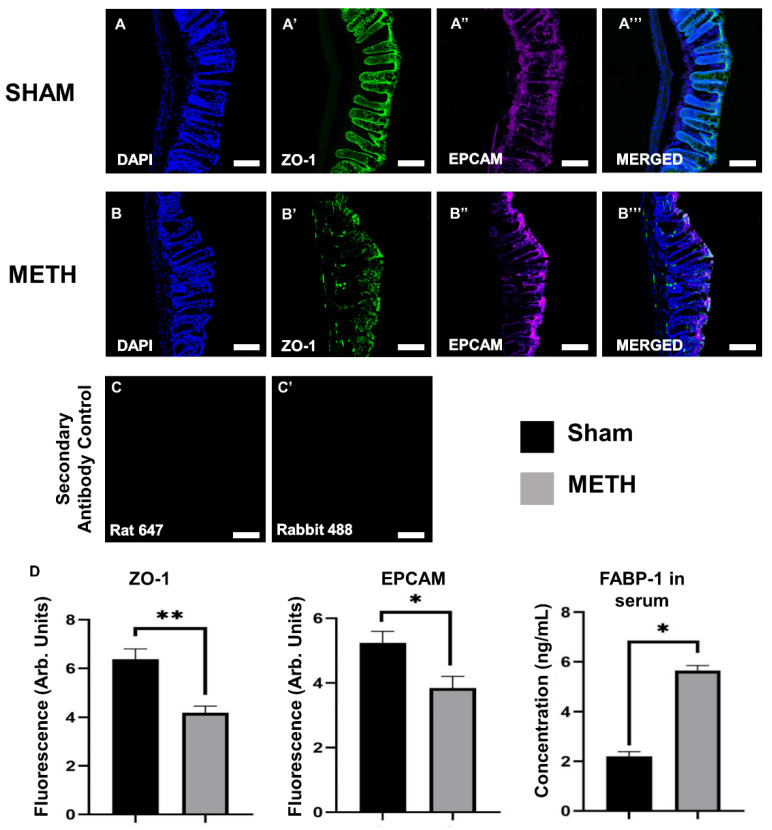
Acute METH exposure increases permeability within the colon. Colon cross-sections of sham-treated (**A**,**A’**,**A’’**,**A’’’**) and METH-treated (**B**,**B’**,**B’’**,**B’’’**) mice. (**A**,**B**) DAPI nuclei staining, (**A’**,**B’**) tight junction proteins labeled with anti- zonula occludents-1 (ZO-1) antibody, (**A”**,**B’’**) epithelial cells labeled with anti-epithelial cell adhesion molecule (EpCAm) antibody, and (**A’’’**,**B’’’**) merged images of DAPI, ZO-1, and EpCAm. (**C**,**C’**) Secondary Antibody Control. To show the non-specific binding of the secondary antibody, the primary antibody was omitted. Colon cross-sections; 20 μm. Images captured at 40×, scale bar: 50 μm. (**D**) Significant differences between sham-treated and METH-treated mice in tight junction permeability, epithelial cells, and concentration of fatty acid-binding protein (FABP)-1 in serum are shown. Results are expressed as mean ± standard error of the mean (SEM), ** p* < 0.05, ** *p* < 0.01.

**Figure 3 ijms-23-11224-f003:**
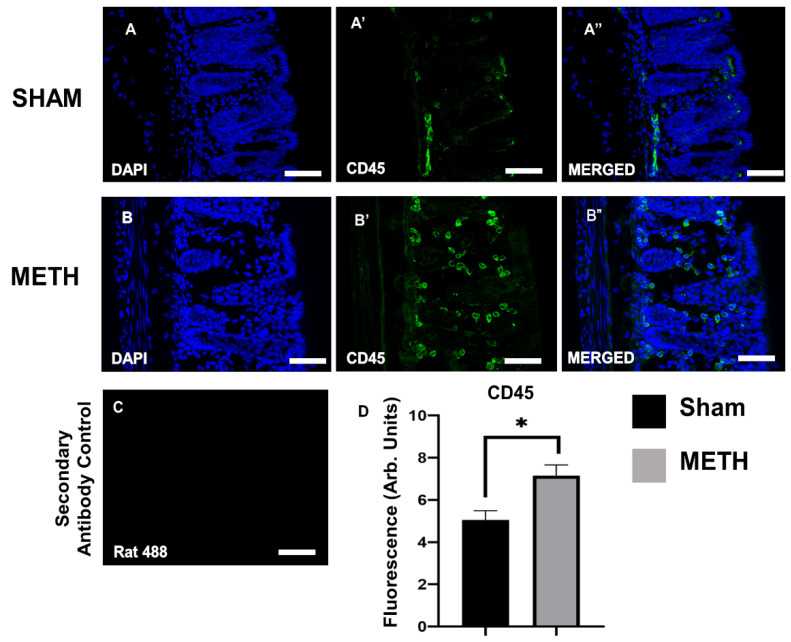
Acute METH exposure induces an inflammatory response within the colon. Colon cross-sections of sham-treated (**A**,**A’**,**A’’**) and METH-treated (**B**,**B’**,**B’’**) mice. (**A**,**B**) DAPI nuclei, (**A’**,**B’**) immune cells labeled with pan-leukocyte anti-CD45 antibody, (**A’’**,**B’’**) merged images of DAPI and CD45. (**C**) The primary antibody was omitted to show the non-specific binding of the secondary antibody. Colon cross-sections; 20 μm. Images captured at 40×, scale bar: 50 μm. (**D**) A significant difference between sham- and METH-treated mice in CD45 expression is noted. Results are expressed as mean ± standard error of the mean (SEM), * *p* < 0.05.

**Figure 4 ijms-23-11224-f004:**
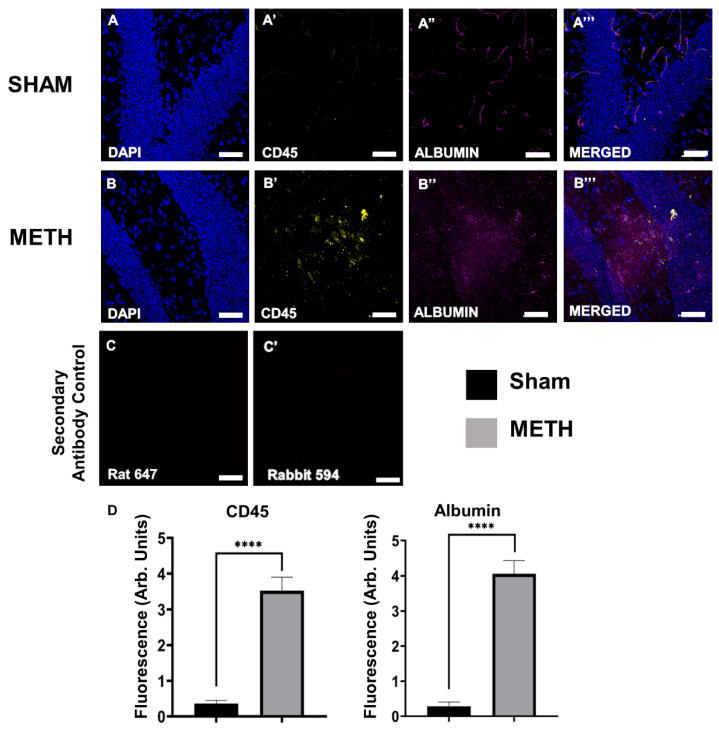
Acute METH exposure induces BBB permeability and inflammatory responses within the brain. Brain cross-sections of sham-treated (**A**,**A’**,**A’’**,**A’’’**) and METH-treated (**B**,**B’**,**B’’**,**B’’’**) mice. (**A**,**B**) DAPI nuclei, (**A’**,**B’**) immune cells labeled with pan-leukocyte marker, anti-CD45 antibody, (**A’’**,**B’’**) gut-derived albumin labeled with an anti-albumin antibody and, (**A’’’**,**B’’’**) merged images of DAPI, CD45, and albumin. (**C**,**C’**) Secondary Antibody Control. The primary antibody was omitted to show the non-specific binding of the secondary antibody. Brain sagittal section level 10–15 using 15 µm intervals. Images captured at 40×, scale bar: 50 μm. (**D**) A significant difference between acute sham-treated and METH-treated mice in CD45 and albumin expression is shown. Results are expressed as mean ± standard error of the mean (SEM), **** *p* < 0.001.

**Figure 5 ijms-23-11224-f005:**
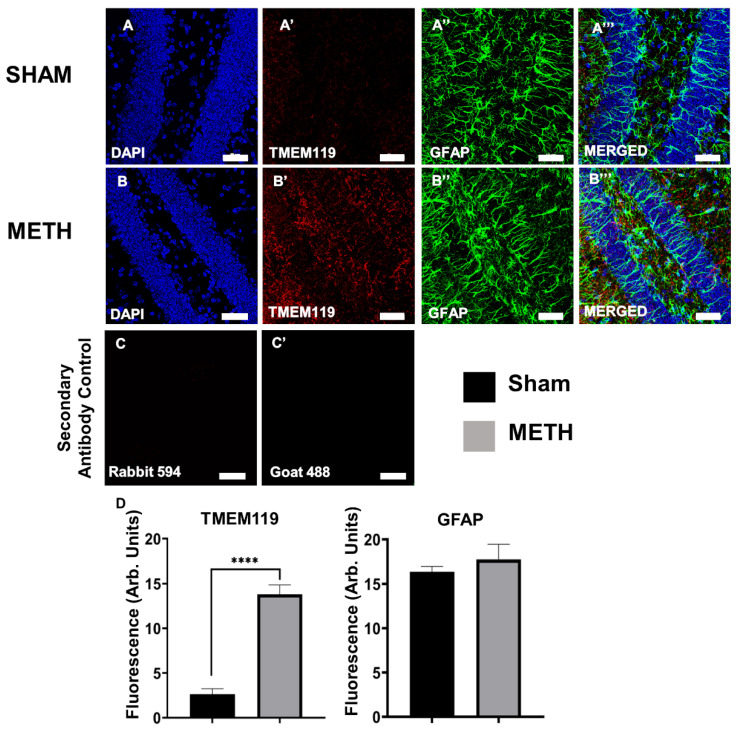
Acute METH exposure induces gliosis within the brain. Brain hippocampus cross-sections of sham-treated (**A**,**A’**,**A’’**,**A’’’**) and METH-treated (**B**,**B’**,**B’’**,**B’’’**) mice. (**A**,**B**) DAPI nuclei staining, (**A’**,**B’**) astrocytes labeled with anti-GFAP antibody, (**A’’**,**B’’**) microglial cells labeled with anti-TMEM119 antibody, and (**A’’’**,**B’’’**) merged images of DAPI, GFAP and TMEM119. Coronal sections; 20 μm; bregma −1.82 mm. Images captured at 40×, scale bar: 50 μm. (**C**,**C’**) Secondary Antibody Control. (**D**) Statistical analysis of Fluorescence (Arb. Units). No significant differences were noted between the sham- and METH-treated mice in GFAP. Significant differences were shown between sham- and METH-treated mice in TMEM 119. Results are expressed as mean ± standard error of the mean (SEM), **** *p* < 0.001.

**Figure 6 ijms-23-11224-f006:**
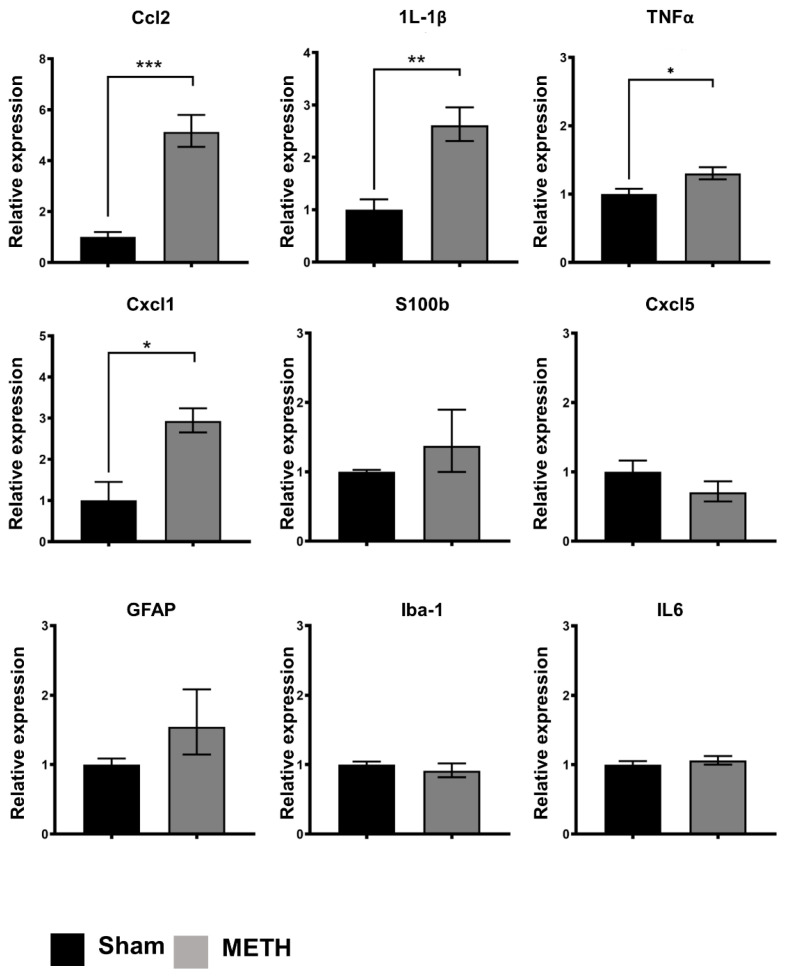
Acute METH exposure induces expression of pro-inflammatory markers in the mid-brain. Pro-inflammatory, chemokine, and glial cell markers expression in the brain. Expression of pro-inflammatory cytokines (IL-1β, TNFα, and IL-6), chemokines (Cxcl1, Ccl2, and Cxcl5), and glial cell markers (GFAP, Iba-1, and S100b) were determined by real-time polymerase chain reaction (RT-PCR) in brain samples for sham- and METH-treated mice. Values are folding increase ± standard error of the mean (SEM) concerning the mean of controls, *** *p* < 0.005, ** *p* < 0.01, * *p* < 0.05.

**Figure 7 ijms-23-11224-f007:**
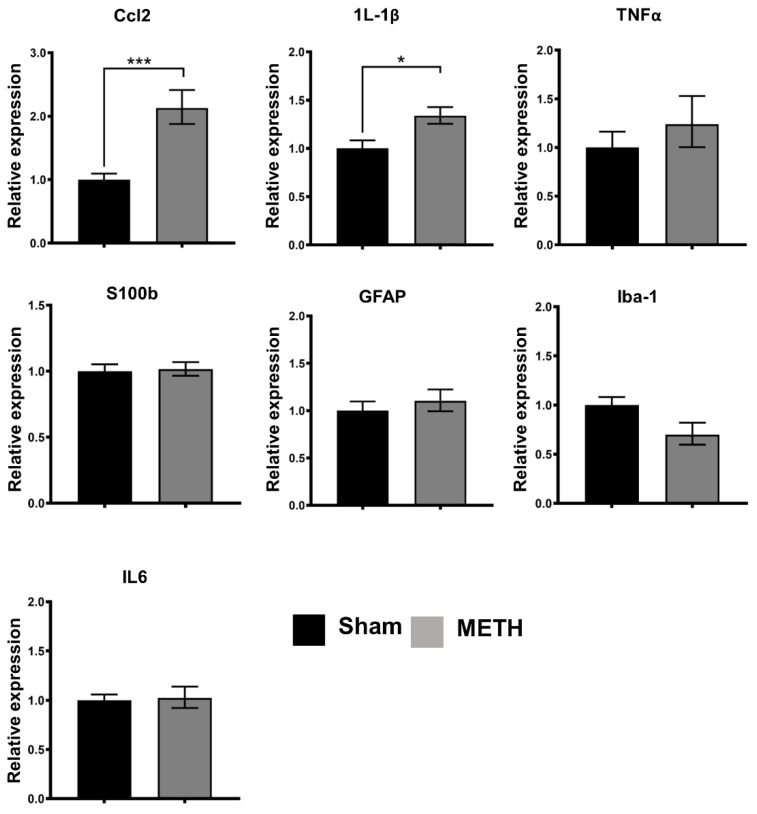
Pro-inflammatory, chemokine, and glial cell markers expression in the hippocampus. Expression of pro-inflammatory cytokines (IL-1β, TNFα, and IL-6), chemokines (Cxcl1, Ccl2, and Cxcl5), and glial cell markers (GFAP, Iba-1, and S100b) determined by real-time polymerase chain reaction (RT-PCR) in the hippocampus of sham- and METH-treated mice. Values are folding increase ± standard error of the mean (SEM) with respect to the mean of controls, *** *p* < 0.005, * *p* < 0.05.

**Figure 8 ijms-23-11224-f008:**
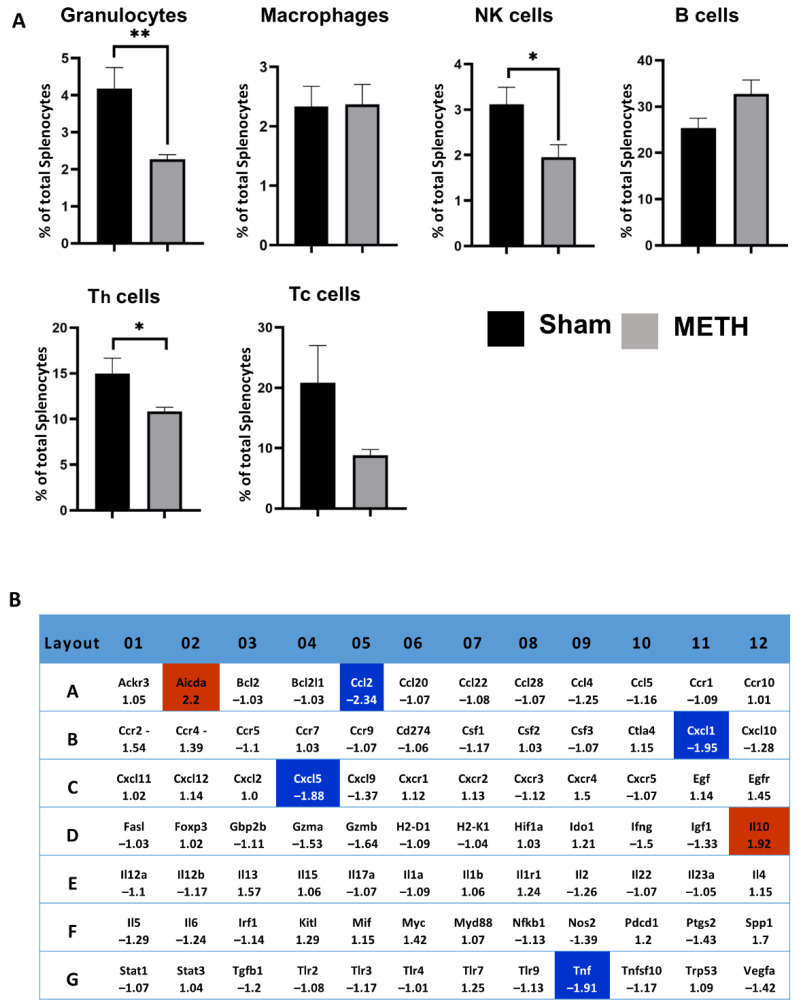
Cellular and gene changes in the spleen. (**A**) Percentage of splenocyte cells (granulocytes, macrophage, natural killer (NK) cells, B cells, helper T (Th) cells, and cytotoxic T (Tc) cells), at time = 3 h following METH administration. Figures show % total splenocyte cells ± standard error of the mean (SEM) with respect to the mean of shams, *** p* < 0.01, ** p* < 0.05. (**B**) RT^2^ Profiler PCR gene array showing upregulated (red) and downregulated (blue) fold-change mRNA gene expression in the spleen of the sham- and METH-treated groups.

**Figure 9 ijms-23-11224-f009:**
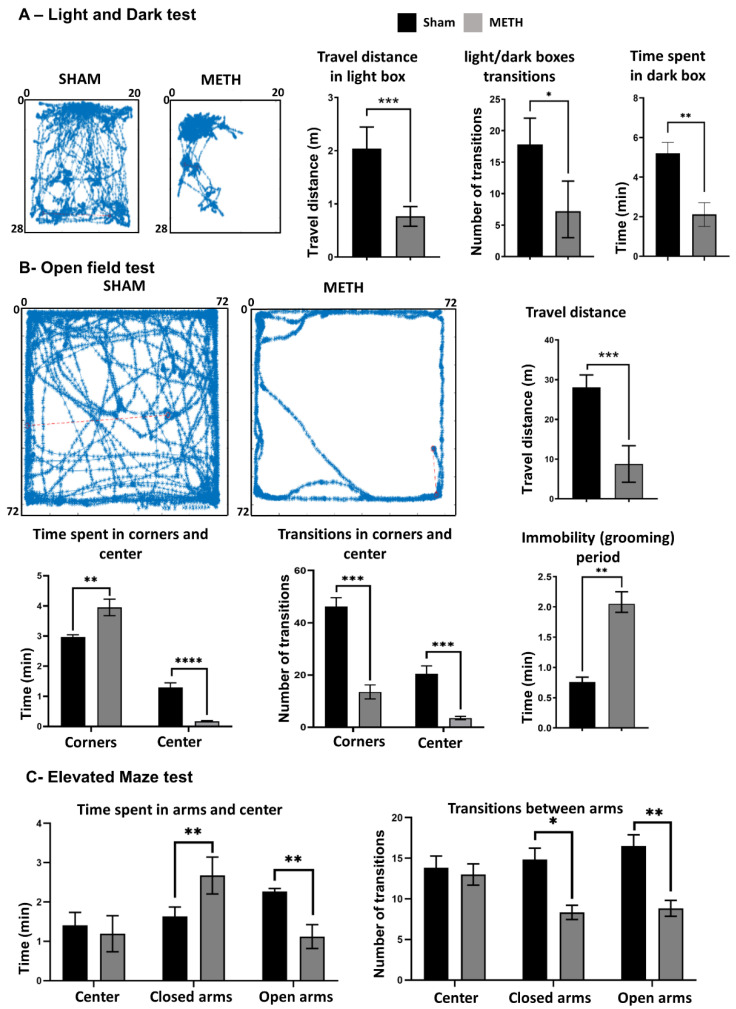
Acute METH exposure induces anxiety. (**A**) Anxiety test, light, and dark test (LDT) results include exploration map in the lightbox, Travel distance in the lightbox (m), Light/dark boxes transitions, and time spent in the dark box; (**B**) Anxiety test, open field test (OFT) results including exploration map in the arena, Travel distance in the arena (m), time spent in corners and center, transition in corners and center, and immobility (grooming) period; (**C**) Anxiety test, elevated maze test (EMT) results including time spent in open and closed arms, and transition between open and closed arms. Results are expressed as mean ± standard error of the mean (SEM). ** p* < 0.05, *** p* < 0.01, ***** 0.001 *< p* < 0.01, **** *p* < 0.001.

## Data Availability

The data presented in this study are available on request from the corresponding author.

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
