# Peer review of "Methamphetamine Induces Systemic Inflammation and Anxiety: The Role of the Gut–Immune–Brain Axis"

_ijms, 2022, doi:10.3390/ijms231911224_

Round 1

Reviewer 1 Report

Mental diseases constitute a major problem of public health that is associated with increased risk of mortality and poor quality of life. Drug abuse and poor quality of life are considered as a major problem that worsens the prognosis of subjects suffering from mental diseases. In this aspect, this text aimed to use methamphetamine (METH) to evaluate the association of intestinal permeability and abuse drug induced mental diseases. In this text, authors expressed the gastrointestinal-immune-nervous systems might play an important role in this animal model.

1.     Authors did conduct the OFT and FST to check drug induced rats with the anxiety-like behaviors, however, such animal behavioral tests presented the various behaviors to mimic the phycological status of mice. The other behavioral tests parameters other than OFT & EPT may necessary to double check the mental status. Furthermore, the differences of behavioral tests should be mentioned in the sections of Introduction and Discussion. Such information should be provided.

2.     Under the drug administration, the bioavailability of METH should be considered. Does gut leaks only found in colon rather than intestine? Such metabolites and biochemical parameters may provide some hints, however, I did not find such information.

3.     The data of enteric microbial toxins such as indoles, secondary bile acids or other microbiota metabolites (LPS) should be provided that might activate immune cells and cause 'leak' from the gut into the bloodstream and travel to the brain. In results of gut microbiota, the information was weak and did not point out the connection of METH and gut microbiota. I did not find any information of that.

4.     Over all, this was an interesting topic but lack of direct evidences to demonstrate the authors’ aim. Authors should provide the solid and direct data to convince reader that METH affects the status of neuropsychiatric behaviors via gut microbiota or it metabolites. Please provide more sensible information. I do not recommend it in current status.

Author Response

Reviewer 1: Mental diseases constitute a major problem of public health that is associated with increased risk of mortality and poor quality of life. Drug abuse and poor quality of life are considered as a major problem that worsens the prognosis of subjects suffering from mental diseases. In this aspect, this text aimed to use methamphetamine (METH) to evaluate the association of intestinal permeability and abuse drug induced mental diseases. In this text, authors expressed the gastrointestinal-immune-nervous systems might play an important role in this animal model. 1. Authors did conduct the OFT and FST to check drug induced rats with the anxiety-like behaviors, however, such animal behavioral tests presented the various behaviors to mimic the phycological status of mice. The other behavioral tests parameters other than OFT & EPT may necessary to double check the mental status. Furthermore, the differences of behavioral tests should be mentioned in the sections of Introduction and Discussion. Such information should be provided. Authors response: The authors appreciate your constructive feedback. In this study three established behavioral tests, including light and dark test (LDT), open field test (OFT), and elevated maze test (EMT), were performed to assess anxiety levels following METH compared to sham administration. In total 11 parameters were considered to evaluate the level of anxiety in a mouse model with all parameters presented in Figure 9. Our findings showed that METH-treated animals have a higher level of anxiety than the sham-treated group. More information regarding the differences between behavioural tests used in this study have been provided in the revised manuscript (lines: 363- 380). 2. Under the drug administration, the bioavailability of METH should be considered. Does gut leaks only found in colon rather than intestine? Such metabolites and biochemical parameters may provide some hints, however, I did not find such information. Authors response: The authors appreciate your feedback. We have provided more information about the bioavailability of METH in the revised manuscript (lines: 454 - 463), which improved the quality of this paper. Thank you. In this study we evaluated the METH-induced changes in tight junction proteins only in the colon. In the future we will evaluate the METH-induced changes in the small intestine as well. We have added more information on the METH-induced changes in the gut permeability to the Conclusions and Future Directions section (lines: 652 - 674). The data of enteric microbial toxins such as indoles, secondary bile acids or other microbiota metabolites (LPS) should be provided that might activate immune cells and cause 'leak' from the gut into the bloodstream and travel to the brain. In results of gut microbiota, the information was weak and did not point out the connection of METH and gut microbiota. I did not find any information of that. Authors response: Thanks for your feedback. We are currently expanding the present study to evaluate the implication of METH-induced changes on the gut microbiota and their metabolites. To prevent any confusion, we have excluded all related discussions on microbiota in the revised version that we hope could improve the quality of the paper. 4. Over all, this was an interesting topic but lack of direct evidences to demonstrate the authors’ aim. Authors should provide the solid and direct data to convince reader that METH affects the status of neuropsychiatric behaviors via gut microbiota or it metabolites. Please provide more sensible information. I do not recommend it in current status. Authors response: Thanks for your feedback. Herein, we examined the complex interaction of the gastrointestinal, immune, and nervous systems following an acute METH dose. Our future study will continue the present study, where we will evaluate the implication of METH-induced changes in the gut microbiota and their association with behavioura changesl. To prevent any confusion, we have excluded all related discussions on microbiota in the revised version and discussed this issue in the Conclusions and Future Directions section (lines: 645 - 722) that we hope could improve the quality of the paper.

Reviewer 2 Report

The manuscript „Methamphetamine induces systemic inflammation and anxiety: The role of the gut-immune-brain axis” presents a broad range of experiments on methamphetamine effects on neural, digestive and immune systems.

1. The effects of METH are discussed in the Introduction. The statement (line 80) “The findings of this study explain the implications of developing neuropsychiatric disorders, including anxiety following METH administration. Then, in the Conclusions: “This study determined the cellular and molecular changes in the gut, brain, and spleen concurrently to elucidate further the neuro-immune interactions that might predispose the development of METH-induced mental health issues.”

The text presents the results of the experiments. Which findings are new, what new connections were found? It seems that a final part of discussion is missing, as the Conclusions are rather general.

With the results presented in the manuscript, the cause-result order may not be completely explained. Is gastrointestinal inflammation primary result of METH use, or general inflammation affects gut as a secondary effect? The study is performed after single dose of METH, please explain the rationale for the acute study versus one representing addiction.

The animals were euthanized after behavior studies. How long after METH injection did this happen? Was the time identical for all animals?

I would suggest moving Figure 9 to the very beginning of results, for a better explanation of experimental design.

2. In line 159, there is information “METH abuse triggers alterations in gut microbiota,  further stimulating an inflammatory response [56], which may explain our findings of increased immune cells within the colon”. The next sentence links microbiome alterations to CNS changes. There are no direct experiments on microbiota, just inflammation markers. I do not think the conclusion in line 161-164 is justified, please explain this in more details.

3. More details on METH BBB permeability should be included in Introduction (time, concentration, half-life/clearance). The link between neural and immune system is established, how the direct action of METH affects the results? As stated in line 184,  “Findings are consistent with previous reports demonstrating an association between METH exposure and increased permeability of the BBB in brain regions, including the hippocampus [61-63].” What new effects/observations does this study provide?

4. Please check the caption for Figure 3, “Acute METH exposure induces brain permeability and inflammatory responses” versus BBB permeability.

5. The Authors do not comment on the difference between their results on GFAP and [64].

6. More detailed information on “Nevertheless, it is clear from previous observations that  METH effects in the brain are region/time/dose-dependent [67].” either in Introduction or in the 2.6 would be beneficial.

7. Figure 5 shows relative results (in 3.9, normalization is described and reference genes named). Please provide reference to this procedure. Folding increase is also not exactly clear.

8. The results from spleen studies indicate immunosuppression. This phenomenon is discussed in 2.7, but not in general conclusion. Did the Authors consider the time factor? How the suppression was assessed as transient (line 290)?

9. Please check lines 332 and 336, they seem to describe the same effect.

10 The description of sample preparation: METH ... “was diluted in 0.1 ml injectable water”. If it was diluted, what was the solvent and concentration of the original solution? If the maximum injection volume did not exceed 200 μl per injection for all mice, and 0.1 ml water was used, does this mean that every injection was prepared separately or doses were combined?

11. Please explain (line 422) Testing was performed after 12 pm.

12. Please explain (line 475) “were embedded in 100% optimal cutting temperature (OCT-compound, Tissue-Tek, USA)”

13. Please provide reference (line 559) (Livak and Schmittgen, 2001).

The Authors mention development of new treatments. Which direction the development should go?

Style and notation comments:

- please use the correct form of IL-1β in the text (line 34), TNFα

- line 214: eryhroid2 – space missing?

- There are different versions of GFAP, please verify the font sizes    

- Line 260: is a recognized sequela following glial activation??

- symbols (multiplication) and test details (for example, two descriptions of Zamboni, with different presentation of picric acid concentration, catalogue number only in line 550, Qiagen with Hilden, Germany or without it) need checking and unification.

- please correct the DAPI name (4’,6’-diamino-2-phenylindole dihydrochloride)

- please check reference 13

- please remove special characters in the title in reference 58

Author Response

Reviewer 2: The manuscript „Methamphetamine induces systemic inflammation and anxiety: The role of the gut-immune-brain axis” presents a broad range of experiments on methamphetamine effects on neural, digestive and immune systems. 1. The effects of METH are discussed in the Introduction. The statement (line 80) “The findings of this study explain the implications of developing neuropsychiatric disorders, including anxiety following METH administration. Then, in the Conclusions: “This study determined the cellular and molecular changes in the gut, brain, and spleen concurrently to elucidate further the neuro-immune interactions that might predispose the development of METH-induced mental health issues.” The text presents the results of the experiments. Which findings are new, what new connections were found? It seems that a final part of discussion is missing, as the Conclusions are rather general. Authors response: The authors appreciate your feedback. We have revised the Conclusions and Future Directions section to address this comment (lines: 645 - 722). With the results presented in the manuscript, the cause-result order may not be completely explained. Is gastrointestinal inflammation primary result of METH use, or general inflammation affects gut as a secondary effect? Authors response: Thanks for your feedback and question. The gastrointestinal inflammation determined by a significant increase in the expression of CD45+ cells was observed only in METH-treated group but not sham-treated group (Figure 3). Therefeore, our results indicate that the inflammatory reaction within the gut is the primary result of METH use. The study is performed after single dose of METH, please explain the rationale for the acute study versus one representing addiction. Authors response: Thanks for your feedback. In this study, we used a single dose of METH as the previous studies showed that a single dose of METH causes neurotoxicity (lines: 454 - 457). We have also established a chronic METH use animal model, the results of the studies in this model will be presented in another paper. The animals were euthanized after behavior studies. How long after METH injection did this happen? Was the time identical for all animals? Authors response: Thanks for your constructive feedback. The experimental time for all animals was identical, exactly 3 hrs after METH administration. The required information has been added to the manuscript (line: 472 - 473). I would suggest moving Figure 9 to the very beginning of results, for a better explanation of experimental design. Authors response: Thanks for your feedback. We have moved Figure 9 to the beginning of the Results section as recommended. 2. In line 159, there is information “METH abuse triggers alterations in gut microbiota, further stimulating an inflammatory response [56], which may explain our findings of increased immune cells within the colon”. The next sentence links microbiome alterations to CNS changes. There are no direct experiments on microbiota, just inflammation markers. I do not think the conclusion in line 161-164 is justified, please explain this in more details. Authors response: Thanks for your constructive feedback. We have revised the manuscript accordingly. The above mentioned lines have been removed and a Future Directions section has been added to the conclusion to include these future studies (lines: 645 - 722). 3. More details on METH BBB permeability should be included in Introduction (time, concentration, half-life/clearance). The link between neural and immune system is established, how the direct action of METH affects the results? As stated in line 184, “Findings are consistent with previous reports demonstrating an association between METH exposure and increased permeability of the BBB in brain regions, including the hippocampus [61-63].” What new effects/observations does this study provide? Authors response: METH increases BBB permeability, inducing damage by altering the structure of proteins involved in BBB stability. METH alters BBB permeability via dysregulation of the tight junction proteins, including occludin and ZO proteins. In vitro studies using a single dose of METH (1µM) on endothelial monocultures showed increased permeability within less than 1 hr. Moreover, a previous study by Matines et al. (2011) reported that a single dose of METH (30 mg/kg) in mice leads to a peak plasma concentration after about 1 hr. Administration of METH at such concentrations in rodents consistently leads to BBB breakdown. In mice brain, significant accumulation of plasma proteins (albumin or IgG) was observed after several hours. In a similar study by Bowyer et al. (2008) findings showed that administration of a single dose of METH (40 mg/kg) in mice induced BBB changes for 1.5 hrs to 3 days. In our study we used a single dose of METH (30 mg/kg), which induced increased BBB permeability evidenced by the presence of gut-derived albumin in the brain tissue. We have added this information to the Results and Discussion section 2.4 (lines: 191 - 204). This study provides novel data on the complex interactions of the gastrointestinal-immune-nervous systems following an acute METH dose administration. Findings present new data on the link between intestinal wall disruption leading to inflammation and the presence of gut-derived albumin suggesting METH-induced blood-brain barrier disruption. We have revised the Conclusions and Future Directions section to address this comment (lines: 645 - 722). 4. Please check the caption for Figure 3, “Acute METH exposure induces brain permeability and inflammatory responses” versus BBB permeability. Authors response: Thanks for your feedback. We have revised the Figure 3 (which is now Figure 4) caption accordingly. 5. The Authors do not comment on the difference between their results on GFAP and [64]. Authors response: Thanks for your constructive feedback. We have added the following information to the manuscript: Our results showed no significant increase in GFAP expression 3 hours after METH administration. This finding is opposite to previously reported results for in vitro studies of rat fetal mesencephalic cell lines where METH increased GFAP expression [Reference 65], which was inhibited by benzamide, an inhibitor of ADP-ribosylation. These differences might be because of differences between in vivo and in vitro models. (lines: 234 - 238). 6. More detailed information on “Nevertheless, it is clear from previous observations that METH effects in the brain are region/time/dose-dependent [67].” either in Introduction or in the 2.6 would be beneficial. Authors response: Thanks for your constructive feedback. The following information regarding METH effects in the brain is provided in the Results and Discussion section 2.6 (lines: 271 - 280): The previous study showed that an acute high dose of METH (30 mg/kg) induces an early increase in the expression levels of IL-6 mRNA in the hippocampus, frontal cortex, and striatum, and TNFα mRNA only in the hippocampus and frontal cortex [REF 68]. Furthermore, a similar study evaluated the effect of a single doses of METH (0.5, 1, 2, and 4 mg/kg) 2 hrs apart in a rat model [Reference 69]. Findings revealed that METH had a dose-dependent stimulatory effect on locomotor activity over the 8 hrs. A significant increase in dopamine concentration was reported in the frontal cortex with the highest dose of METH 2 h after the dose administration [Reference 69]. This effect was dose- and region-specific, as no significant changes were reported for lower doses, nor was a significant change reported for other brain regions. 7. Figure 5 shows relative results (in 3.9, normalization is described and reference genes named). Please provide reference to this procedure. Folding increase is also not exactly clear. Authors response: All RT-PCR data were normalised to the house keeping genes. This has been made clearer in the corresponding methods section 3.9 (lines 630-634): Normalized relative quantities (NRQ) were determined using the formula 2-delta-Ct, where delta-Ct represents the difference between the METH- and sham-treated groups, normalized to the mean of stable reference genes (hippocampus Actin, Gus, Gapdh; mid- brain Gapdh, Gus). Data is presented as fold-change, compared to non-treated controls. The information regarding fold change is provided in the methods section (lines 615-619): Fold change was determined using the delta-delta CT method. The arrays were performed in duplicate, and specific genes showing an expression difference of ≥ 2-fold relative to control were selected for validation using gene-specific RT-qPCR. Low genes expressed at levels near the detection limits (Ct > 35) and genes without a unique melting point were not considered further. 8. The results from spleen studies indicate immunosuppression. This phenomenon is discussed in 2.7, but not in general conclusion. Did the Authors consider the time factor? How the suppression was assessed as transient (line 290)? Authors response: Thanks for your feedback. The changes in the spleen are at one time point, 3 hours post METH administration. We did determine immune cell changes at time 24 hours, and cells were back to baseline levels, hence the effect was transient. We have added further information in the text (lines 331-332). The immunosuppression in the spleen has been added to the Conclusions and Future Directions section (lines 693-696). 9. Please check lines 332 and 336, they seem to describe the same effect. Authors response: Thanks for your feedback. The first sentence describes the results of the Light and dark test, and the second sentence describes the open field test results. Both tests confirm that mice have decreased activity and a reduction in exploratory and locomotor behavior, indicating a higher level of anxiety. 10 The description of sample preparation: METH ... “was diluted in 0.1 ml injectable water”. If it was diluted, what was the solvent and concentration of the original solution? If the maximum injection volume did not exceed 200 μl per injection for all mice, and 0.1 ml water was used, does this mean that every injection was prepared separately or doses were combined? Authors response: Thanks for your question. We prepared all injections (working solution) separately based on animal body weight to make sure all animals would receive a consistent dose of 30 mg/kg body weight of METH. The solvent was sterile saline. We have added this information to the Methods section (lines 450-454). 11. Please explain (line 422) Testing was performed after 12 pm. Authors response: Thanks for your constructive feedback. We have deleted this sentence and added “The experimental time for all animals was identical, exactly 3 hrs after METH administration.” (lines: 472 - 473), to clarify this issue. 12. Please explain (line 475) “were embedded in 100% optimal cutting temperature (OCT-compound, Tissue-Tek, USA)” Authors response: Thanks for your feedback. We have edited the sentence to read “After washing, tissues were embedded in optimal cutting temperature compound (OCT-compound, Tissue-Tek, USA) and frozen using liquid-nitrogen and 2-methyl butane (Sigma-Aldrich, Australia) and stored in a -80 °C freezer as in our previous studies [Ref 100]”. This is a standard procedure used for embedding and freezing tissues for cryostat sectioning and immunohistochemistry. 13. Please provide reference (line 559) (Livak and Schmittgen, 2001). Authors response: Thanks for your feedback. The required reference has been added to the manuscript: reference [102] (line 615). The Authors mention development of new treatments. Which direction the development should go? Authors response: Thanks for your constructive feedback. We have rewritten the conclusion section and added the following information to this section: “The findings of this study suggest several potential therapeutic pathways that might lead to designing an efficient treatment against METH-induced anxiety and depression. These options include but are not limited to (i) targeting leaky gut to prevent entrance of endotoxins to the blood circulation and brain, (ii) reducing BBB disruption, (iii) immunomodulation to prevent inflammation.” (lines: 715-720). Style and notation comments: - please use the correct form of IL-1β in the text (line 34), TNFα - line 214: eryhroid2 – space missing? - There are different versions of GFAP, please verify the font sizes - Line 260: is a recognized sequela following glial activation?? - symbols (multiplication) and test details (for example, two descriptions of Zamboni, with different presentation of picric acid concentration, catalogue number only in line 550, Qiagen with Hilden, Germany or without it) need checking and unification. - please correct the DAPI name (4’,6’-diamino-2-phenylindole dihydrochloride) - please check reference 13 - please remove special characters in the title in reference 58 Authors response: Thanks for your constructive feedback. We have made all required corrections in the revised version.

Reviewer 3 Report

This is an interesting study which examines the effect of Meth treatment in mice on gut leakiness, inflammation, blood brain barrier leakage and gliosis/neuroinflammation.

Please see my comments/questions below:

1. Apart from the presence of FABP-1 in the circulation what other evidence of increased gut leakage was examined and how can the authors specifically conclude that FABP-1 was solely of gut origin and nor from the liver? Was evidence of liver damage investigated in the mouse model?

2. How was the dose of Meth which the animals were treated with derived and does it make the levels in human addicts?

3. The presence of increase albumin in the hippocampus was used a evidence of increased blood brain barrier leakage, but shouldn`t the levels of of NF-L and CHI3L1 also have been assessed?

4. The presence of TMEM119 protein in the brain hippocampus wa used to indicate evidence of gliosis but other markers should have been used to support this assumption.

5. At present the results of this study cannot really conclude concrete evidence of neuroinflammation since there wasn`t  an increase in the gene expression of in the classical microglial marker Iba-1 or astrocyte markers S100b and GFap. Furthermore,  was evidence of inflammation or increased oxidative stress assessed rather than solely relying on gene expression?

6. The abstract provides no actual details of the aims of the study or that this is an investigation in mice. An aims should also be present at the end of the introduction.

7. The conclusion is a little oblique and requires comments about the actual findings of the investigation.

Author Response

Reviewer 3: This is an interesting study which examines the effect of Meth treatment in mice on gut leakiness, inflammation, blood brain barrier leakage and gliosis/neuroinflammation. Please see my comments/questions below: 1. Apart from the presence of FABP-1 in the circulation what other evidence of increased gut leakage was examined and how can the authors specifically conclude that FABP-1 was solely of gut origin and nor from the liver? Was evidence of liver damage investigated in the mouse model? Authors response: Thanks for your comment. In this study we employed immunohistochemistry in intestinal tissues to understand the influence of METH administration on tight junction proteins compared to sham-treated groups. Figure 2 presents the expression of tight junction protein, ZO-1, and epithelial cell adhesion molecule, EpCAm, in colon tissues for METH- and sham-treated mice. METH administration causes a significant reduction in the expression of tight junction proteins within the intestine. Reduced ZO-1 and EpCAM expression indicates tight junction disruption, leading to increased intestinal permeability and gastrointestinal inflammation. In this study we measured FABP-1 levels using Quantikine ELISA kit (mouse/rat FABP1/LFABP) (R&D Systems, in vitro Technologies) which measures FABP from both the intestine and liver. In the future we will measure IFABP, which is more specific to intestinal FABP. We have provided more information on different types of FABP in the manuscript (lines: 583 - 587). 2. How was the dose of Meth which the animals were treated with derived and does it make the levels in human addicts? Authors response: Thanks for your question. We prepared all injections (working solution) separately based on animal body weight to make sure all animals would receive a consistent dose of 30 mg/kg body weight of METH. This study followed the simple practice guide for dose conversion between animals and humans [DOI: doi: 10.4103/0976-0105.177703] to calculate the METH dose similar to the drug dose in human users. We have provided the required information and references in the revised version (lines: 451 - 463). 3. The presence of increase albumin in the hippocampus was used a evidence of increased blood brain barrier leakage, but shouldn`t the levels of NF-L and CHI3L1 also have been assessed? Authors response: The authors appreciate your constructive feedback. We will certainly consider the levels of NF-L and CHI3L1 in our further studies. We have added the following to the Conclusions and Future Directions section: “METH decreases BBB structural proteins and increases BBB permeability to various molecules and gut-derived endotoxins into the brain. In this study presence of gut-derived albumin was used to confirm increases in the BBB permeability. However, the levels of neurofilament light protein (NFL), a cytoskeletal polypeptide of the axon, and chitinase 3-like 1 (CHI3L1, also known as YKL40 or gp39), a glycoprotein secreted by activated glia in the CNS, could be considered to confirm BBB leakage in future studies.” (lines: 689 - 692). 4. The presence of TMEM119 protein in the brain hippocampus was used to indicate evidence of gliosis but other markers should have been used to support this assumption. Authors response: The authors appreciate your constructive feedback. We will certainly consider other markers in our further studies. 5. At present the results of this study cannot really conclude concrete evidence of neuroinflammation since there wasn`t an increase in the gene expression of in the classical microglial marker Iba-1 or astrocyte markers S100b and GFap. Furthermore, was evidence of inflammation or increased oxidative stress assessed rather than solely relying on gene expression? Authors response: We appreciate your constructive feedback. However, it is clear from the current studies that acute METH induces an upregulation of Ccl2, IL-1b, TNFa and Cxcl1 inflammatory markers in mid-brain, as well as an increase of Ccl2, IL-1b in the hippocampus. We will consider further detailed inflammatory markes in the brain in our future studies. 6. The abstract provides no actual details of the aims of the study or that this is an investigation in mice. An aims should also be present at the end of the introduction. Authors response: Thanks for your comment. We have revised the abstract accordingly, which we hope could improve the quality of the paper (lines: 26 - 44). 7. The conclusion is a little oblique and requires comments about the actual findings of the investigation. Authors response: Thanks for your comment. We have revised the conclusion section accordingly, which we hope could improve the quality of the paper (lines: 645 - 722).

Round 2

Reviewer 1 Report

Authors replied to the considerations properly. I have no further questions